# Clinical Evaluation of a New Electronic Periodontal Probe: A Randomized Controlled Clinical Trial

**DOI:** 10.3390/diagnostics12010042

**Published:** 2021-12-25

**Authors:** Oliver Laugisch, Thorsten M. Auschill, Christian Heumann, Anton Sculean, Nicole B. Arweiler

**Affiliations:** 1Department of Periodontology and Peri-Implant Diseases, Philipps-University, 35037 Marburg, Germany; Oliver.Laugisch@uni-marburg.de (O.L.); Auschill@med.uni-marburg.de (T.M.A.); 2Department of Statistics, Ludwig-Maximilians-University Munich, 80539 Munich, Germany; Christian.Heumann@stat.uni-muenchen.de; 3Department of Periodontology, School of Dental Medicine, University of Bern, 3010 Bern, Switzerland; anton.sculean@zmk.unibe.ch

**Keywords:** periodontal disease, periodontitis, probing pocket depth, periodontal diagnostics, clinical attachment loss

## Abstract

Precise measurements of periodontal parameters (such as pocket depths: PPD, gingival margins: GM) are important for diagnosis of periodontal disease and its treatment. Most examiners use manual millimeter-scaled probes, dependent on adequate pressure and correct readouts. Electronic probes aim to objectify and facilitate the diagnostic process. This randomized controlled trial compared measurements of a standard manual (MP) with those of an electronic pressure-sensitive periodontal probe (EP) and its influence on patients’ acceptance and practicability. In 20 patients (2436 measuring points) PPD and GM were measured either with MP or EP by professionals with different levels of experience: dentist (10 patients), 7th and 10th semester dental students (5 patients each). Time needed was measured in minutes and patients’ subjective pain was evaluated by visual analogue scale. Differences were analyzed using the generalized estimating equations approach (GEE) and paired Wilcoxon tests. Mean PPD varied with ΔPPD 0.38 mm between both probes, which was significant (*p* < 0.001), but GM did not (ΔREC 0.07 mm, *p* = 0.197). There was a statistically significant correlation of both probes (Spearman’s rho correlation coefficient GM: 0.674, PPD: 0.685). Differences can be considered robust (no deviation in either direction). The comparison of time needed and pain sensitivity did not result in statistically significant differences (*p* > 0.05).

## 1. Introduction

Periodontal disease is a widespread infectious disease of tooth supporting tissues with a prevalence of more than 50% [1]. It causes a significant economic burden both in the US and in Europe [2]. Chronic inflammation with progressive alveolar bone loss [3] does not only result in tooth-loss [4], there are also clinical and inflammatory relationships with other chronic metabolic, inflammatory, and vascular diseases, such as diabetes [5], cardiovascular diseases [6], chronic obstruction pulmonary disease (COPD) [7], metabolic syndrome, and obesity [8,9]. Therefore, early diagnosis and prevention gains more and more importance.

According to the new classification of periodontal diseases [10], as well as the EFP S3 level guideline for its treatment [11], exact measurements of periodontal parameters are necessary for adequate diagnostics, monitoring treatment-outcomes and the decision of further treatment-options. Modern diagnostic tools, such as prognostic biomarkers like Matrix-Metallo-Proteinase(MMP)-8 extent and improve the information provided by the standard clinical measures since they predict the risk for on-going and future periodontal breakdown by measuring inflammatory mediators [12]. In particular, grading (the prognostic progress) in the framework of the new classification system of both periodontitis [13] and periimplantitis [14] could benefit from biomarkers. Thus, only in-time diagnosis followed by adequate therapy can guarantee effective treatment [15].

However, so far, staging (the severity and extent of the present periodontal breakdown) is dominated by classical measurements, taken by a periodontal probe and radiographs.

Probing pocket depth (PPD), defined as the distance between the gingival margin and the bottom of the periodontal pocket, together with clinical attachment level-loss (CAL-loss) and bone-loss, as well as bleeding on probing (BOP) are clinical signs of periodontal destruction [16]. Thus, PPD measurement is still a fundamental prerequisite for diagnostics, since it differentiates between the healthy and clinically diseased pocket [17,18]. Up to now, manual probing is the ‘golden standard’ even if this analogue, minimal-invasive method has certain limitations due to inter-individual varying pressures while inserting the probe, different inflammatory conditions of the gingival tissue, wrong angulation of the probe, as well as errors while reading the scale or transferring measurements in the dental record [19,20].

In order to minimize errors resulting from differing pressure-forces, periodontal probes with a pressure calibration of 20 g/0.2 Newton have been developed [21]. However, they are very susceptible to defects, much more expensive, and their hygienic preparation is very difficult. First, electronic pressure sensitive probes did not bring any advantage, as they coronally penetrate the junctional epithelium [22] and patients’ acceptance was low.

In recent years, a new, computer-based, electronic and pressure-calibrated probe (PA-ON Parometer, orangedental, Biberach, Germany) has been introduced to the market (Figure 1). Measurements are taken calculating the insertion depth of a thin steel tip with a plastic cover at 20 g/0.2 Newton pressure using resistance. Due to a computer-based measuring-sequence, tooth-to-tooth data is recorded on a chip, read out loud, and transferred wireless in a computer based periodontal-chart.

It was the aim of this randomized, controlled, clinical trial to compare this electronic with a conventional and established manual probe on gaining reliable data and its accordance (primary outcome). In a sub-analysis, it should be examined if clinical experience of the examiner had an influence on measurements. Time needed for data assembling, as well as patients’ acceptance while probing were further outcomes.

It is hypothesized that periodontal diagnostics by electronic is at least comparable to established manual probing. Furthermore, it was questioned if electronic probing could be a faster alternative and being preferred by patients.

## 2. Materials and Methods

### 2.1. Study Center and Subjects

In total, 20 subjects were selected out of the pool of patients of the Department of Periodontology and Peri-Implant Diseases scheduled for reevaluation after systemic periodontal therapy or maintenance care (SPT).

### 2.2. Sample Size Calculation

A sample size calculation was performed in order to compare two methods. Within 95% of limits of agreement and a precision of +/−0.25 mm, a sample size of *n* ≥ (1.96 * sqrt(3) * 1.2/0.25)^2 = 265.53 was calculated. This would make 266 sites, correlating with about 44 teeth and 2 patients. Since patient was the statistical unit, 10 patients per profile of the examiner (with experience, with less experience) should be included.

### 2.3. Study Design and Procedure

The study was structured in accordance with the CONSORT guidelines (http://consort-statement.org/; accessed on 20 December 2021), and registered at the German clinical trial register (DRKS00023738). It has been conducted in full accordance with ethical principle, including the World Medical Association Declaration of Helsinki (version 2008) and was reviewed and approved by the local Medical Ethics Committee (#81/13). The STROBE Statement-checklist was strictly followed. A flow-chart can be seen in Figure 2. Upon written informed consent 20 patients were included in this study. At all teeth, participants received a 6-point per tooth measurement (disto-buccal, buccal, mesio-buccal, disto-oral, oral, meso-oral), resulting in 2436 test-sites. The same measurements either using manual (MP) or electronic probing (EP) had been performed in each participant by the same examiner, while the sequence of measurements had been determined by a computer-based randomization scheme.

A manual periodontal probe with an elasticity of 0.2 N and a millimeter-scale (MP, DB76R, Aesculap, Tuttlingen, Germany) was used to compare measurements with the PA-ON Parometer (EP, orangedental, Biberach, Germany) (Figure 1). Both measurements were performed by a qualified dentist (10 patients), as well as undergraduate students (7th semester: ST7 and 10th semester: ST10; 5 patients and 5 students each) to test potential differences in experience.

The dentist and the students were instructed in using the EP in a training course of two hours including probing on periodontal disease models. All students were already familiar with periodontal measurements with MP.

### 2.4. Clinical Parameters

In each site probing pocket depths (PPD) defined as the distance between the gingival margin and the bottom of the periodontal pocket were measured. Additionally, the position of gingival margins (GM) in relation to the cemento–enamel junction (minus values for recessions, plus values for hyperplasia) were determined in order to complete a whole periodontal chart of each patient. 

### 2.5. Further Parameters

Time needed to complete periodontal charting was counted for each method and each examiner and measured in minutes using a stopwatch;Questionnaires by means of visual analogue scale (VAS) were used to gain information concerning patient’s pain-sensation (0 [no pain] to 10 [extreme pain]) during probing;The subjective experience of the patients was determined by an independently developed questionnaire in the style of Quality of life (QoL);An evaluation by dental professionals was performed in order to determine expectations and satisfaction (VAS) with MP compared to EP.

### 2.6. Data Management/Data Collection Forms

The whole dataset was pseudonymized, stored and transferred to the statistician (CH) for further analysis.

### 2.7. Statistical Analysis

Statistical analysis was carried out using SPSS Version 23 (IBM, Armonk, NY, USA). Data of both measurements (MP/EP) were checked for differences with Bland–Altman Plots. Following descriptive statistics and coefficients of correlation (Spearman’s rho), *p*-values have been determined (1) by a generalized estimating equations approach (GEE) for patients as units of measurement and sites as repeated, possibly correlated, measures, as well as type of probing (MP or EP) as variable of influence or (2) by paired Wilcoxon tests. Statistical significance was set as *p* < 0.05.

## 3. Results

### 3.1. Measurements of Probing Pocket Depths (PPD) and Gingival Margins (GM)

In all examiners, mean PPD varied significantly between MP and EP with a mean ΔPPD of 0.38 mm (*p* < 0.001), while GM revealed only a ΔGM of 0.07 (*p* = 0.195). However, both probes resulted in a high correlation in both parameters. Spearman’s rho for PPD was 0.685 and for GM: 0.674, both being statistically significant (*p* < 0.001) (Table 1).

A subanalysis differentiating between the dentist, 7th and 10th semester dentistry students (ST7, ST10) revealed measurement deviations for both probes (Table 2 and Table 3, Figure 3). A ΔPPD of 0.21 mm in dentist (*p* = 0.002) and of 0.82 mm in ST10 (*p* < 0.001) was significantly different, while a ΔPPD of 0.44 mm in ST7 was not. The measurement of GM showed a ΔGM of 0.07 mm in the dentist and ST7, which was only significant in the dentist (*p* = 0.047) and a ΔGM of 0.23 mm in ST10 which did not reach the level of significance (*p* = 0.203) (Table 2 and Figure 3). Spearman’s rho reached the level of significance in all parameters and examiners. The percentage distribution of the absolute deviation of both (PPD/GM) in repeated measurements per examiner are presented in Table 3. The intra-reader reliabilities (calculating the intra-class correlation coefficient) for the examiners (Dentist, ST10, ST7) for PPD were 0.844, 0.544 and 0.440 and for GM 0.604, 0.729, and 0.572, respectively.

### 3.2. Time Needed for Measurements

The absolute EP measuring time (23 ± 11 min) was in average 2 min faster, but was not a significant timesaver compared with MP (21 ± 11 min) (*p* < 0.05). Time needed decreased with examiners experience (Table 4).

### 3.3. Patients Pain Sensitivity (VAS)

The comparison of VAS pain did not result in statistically significant differences neither dependent on examiners experience nor on probe used (MP: 3.61 ± 2.04, EP: 3.31 ± 2.66) (Table 5).

### 3.4. Patients’ Subjective Experience

Nearly all participants (19 of 20) had not been familiar with EP before and their expectations varied from easier (8), more pleasant (10), more precise (5) to no expectations (7). Although the evaluation of EP in comparison to MP revealed ‘much better’ (4), ‘better’ (6) ‘worse’ (5), and ‘no difference’ (6), most of them would prefer EP (12 EP; 6 MP) for future measurements.

### 3.5. Evaluation by Dental Professionals

All examiners (ST7, ST10, dentist) considered EP to be faster, more precise, and easier. The VAS-values for satisfaction with the clinical use (1 = very satisfied and 10 = not satisfied) showed a moderate satisfaction in dentists (5.11), in ST7 (5.54), while ST10 graded 3.06.

## 4. Discussion

This randomized controlled clinical trial showed that periodontal diagnostics using a new electronic probe can be seen as an alternative to common manual probing.

The new classification of periodontal diseases [10] and new guidelines for its treatment [11] strongly focuses on diagnostics. Therefore, the use of periodontal probes is essential. Tools to facilitate this process by gaining reliable data are necessary.

In each patient the same measurements either using a millimeter-scaled manual (MP) or electronic probe (EP) had been performed by the same examiner and the sequence of measurements had been determined by randomization in order to minimize differences due to probing a second time.

Oringer et al. [23] compared measurements of one single examiner with a manual and an automatic probe in untreated periodontitis patients at different time-points. More variability was documented with manual probing at two follow-up measurements, which supports the measurement at one time-point. Using a millimeter scaled probe has shown that the scale of a probing tip has a statistically significant influence on gaining reliable results and a millimeter scale being more accurate [24,25]. To further improve mechanical measurements and for a proper comparison with the electronic probe a pressure sensitive manual probe was used. It has been shown that constant-force probing provides reproducible results [25,26].

Although it is known that a calibration of examiners leads to reproducible probing [25], the present study design did not implement a calibration of examiners as one patient was examined by only one examiner and it was the aim to compare two periodontal probes by examiners at different skill levels.

The present study revealed a robust correlation of both probes, although a difference of 0.38 mm in PPD was found to be significant. A significant deviation to manual probing could devalue the electronic probe. There are, however, many aspects that should be considered: (1) the overall correlation of both measurements (Spearman’s rho: 0.685 *) was high and statistically significant. This indicates that deviations did not vary in both directions (higher and lower) but were always very similar in one direction. Paraphrased, a mean difference between both probes could be zero, but could be the result of different errors of, e.g., 1 mm in both directions that neutralize each other; (2) reading errors, different angles, as well as anatomical depth of the pocket, severity of the disease, and smoking habits have an impact on probing independent on whether a manual or electronic probe is used [27,28]; (3) values are mostly rounded to the next millimeter while the electronic probe rounds exactly and in both directions.

Thus, the resulting difference in PPD of less than 0.5 mm is in line with the measuring error of ±1 mm for clinical reproducibility [19,29,30].

Overall there were no statistically significant differences in GM measurements using MP and EP. The dentist and ST7 revealed the same divergence of 0.07 mm, however, only in the dentist’s measurements the level of statistical significance was reached. As mentioned before, this difference is also clinically neglectable.

By taking a closer look at the examiners’ experience (with regard to a subanalysis) differences in PPD were still significant in the dentist and in 10th semester students, but not within 7th semester students. Generally, it is assumed that the experience of an examiner has an impact on probing measurements for the diagnosis of periodontal disease [25,31]. It is known that 1 year-training/experience significantly declines former discrepancies both in trained faculty staff and students [31]. It is an interesting finding of the present study, that deviations of ST7 and the dentist were very similar (albeit ST7 needed more time) and always lower than discrepancies of ST10. Within our academic setting 7th semester correlates with ½ year training. However, it can also be speculated that 7th semester students probe more thoroughly and therewith compensate less experience than 10th semester students. It should also be mentioned that ST10 had a 1-year break with no periodontal training after their 7th semester. However, as mentioned before, all discrepancies were lower than the measuring error for clinical reproducibility [19,29,30].

Only one similar study analyzing the same electronic probe was published in 2016 [32]. In a private practice setting, two male examiners compared it with a millimeter-scaled MP in 25 periodontitis patients (3606 measuring sites in 601 teeth). Although both examiners underwent a calibration process, higher PPD and VAS-pain was documented when using MP. Additionally, EP was no statistically significant time-saver for the documentation of PPD.

The present study was performed in a specialized periodontal department and included nearby an equally-sized group of patients. As the study investigated the output of differently qualified examiners, a preceding calibration was not reasonable. Bleeding on probing (BOP) can be a result of too much pressure, but was generally not included into the study setting.

In 1996, Mayfield and co-workers analyzed the precision of 4 different periodontal probes—two manual (one of them with a standardized pressure of 0.20 N) and two electronic ones (one with a standardized pressure of 0.25 N and the other one with a controlled pressure depending on pocket depth) [33]. Although the standard manual probe revealed the deepest measures compared to all pressure-controlled measurements, all correlation-coefficients were located around 0.8 and thereby (including standard deviations) very similar. These results could also be influenced by a lack of familiarity with new techniques in the late 90 ties.

The validity of a constant force electronic probe and conventional probing in comparison within a post-extraction measurement was investigated by Hull and co-workers [34]. The electronic probe reported a PPD difference of 0.48 mm (*p* < 0.01) and the conventional of 0.08 mm (*p* > 0.05), respectively. However, as mentioned before, the difference of 0.48 mm is clinically neglectable.

Perry et al. [35] compared conventional probing with a manual pressure regulated probe and two different electronic ones. Conventional probing resulted in the deepest measurements (*p* < 0.0001), followed by manual pressure regulated (*p* < 0.001). Both electronic probes dominated with fastest measurements and an 90% inter-examiner agreement. Unfortunately, patients’ acceptance was less comfortable (examined by VAS) as manual and conventional probing. This is in contrast to the results of the present study, where patients felt no differences in measurement modalities concerning pain or inconvenience. It might be explicable with technical progress within the last 20 years, especially with respect to measuring tips, now being slimmer and more flexible. When comparing the pure time needed for measuring in MP and EP, no statistically significant difference was found. However, electronic probing with the probe being investigated in this study yields a visible periodontal chart to be used digitally or printed out. In contrast, a dental assistant is necessary for documentation when using a conventional manual probe while the EP captures the measurements, speaks them out loud and creates a full periodontal chart with the corresponding software. This is a timesaver beyond the assessed parameter “time need”, thus being a meaningful alternative to manual probing—especially for dental hygienists that normally work alone.

Although Computer-Tomographic (CT)-scans or digital scanning could be an objective alternative for periodontal measurements, this method is used so far only for soft tissue measurements with respect to recessions and papilla height, whereas pockets cannot be captured. First, studies determined differences of more than 2 mm in 49.5% of measurements compared to manual probing [36]. A further study compared intraoral measurements by a periodontal probe and by a caliper on dental casts with scans of casts in two follow up measurements. Digital measurements were found to improve reproducibility and to lower the variance of measurements within one individual and between different investigators [37]. Despite the fact that up to now many deficits, computer-based techniques are surely an interesting field of research in future since measuring errors relating to pockets could have a significant impact on therapeutic modalities.

## 5. Conclusions

Standard manual probing and the use of an electronic probe correlate very well, albeit PPD measurements showed a statistically significant difference, which was below a known measuring error of 1 mm. Duration for pure measurements and patient acceptance were comparable in both probes. Considering subjective inter-examiner reading errors using a manual probe, objective electronic probing on basis of a standardized resistance can be seen as a practical alternative as the need for assistance and time for documentation is saved.

## Figures and Tables

**Figure 1 diagnostics-12-00042-f001:**
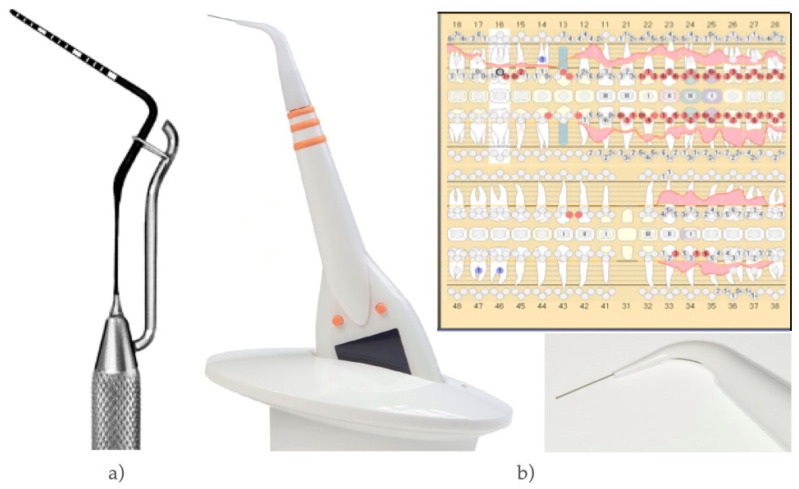
(**a**) Pressure calibrated manual probe (MP); (**b**) Electronic probe (EP), magnification of its disposable tip and periodontal chart (automatically generated).

**Figure 2 diagnostics-12-00042-f002:**
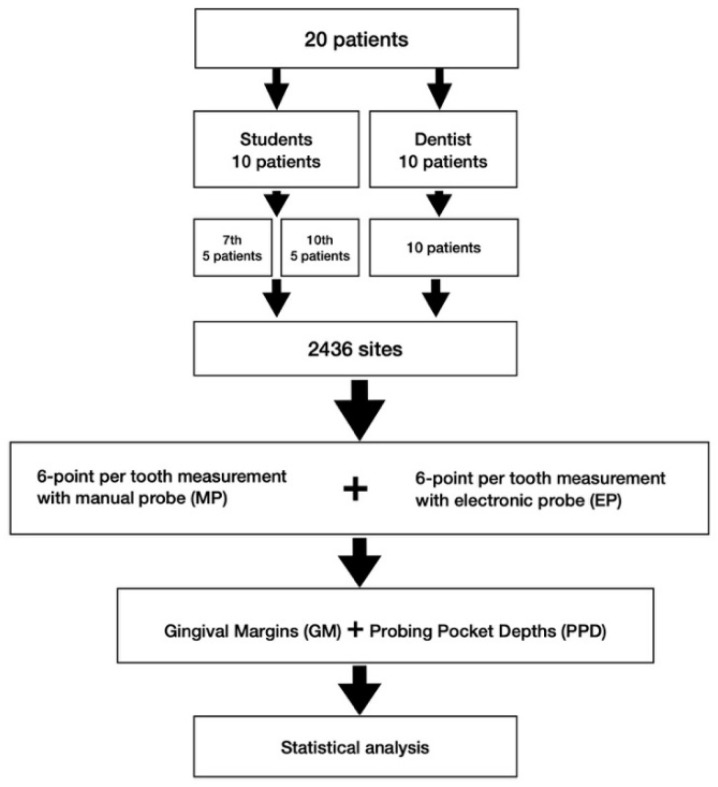
Flow-chart of the study.

**Figure 3 diagnostics-12-00042-f003:**
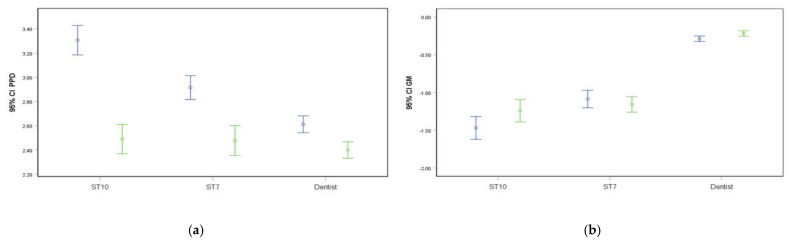
(**a**) PPD for manual (MP; blue) or electronic probing (EP; green) differentiated by level of examiner’s experience (dentist, 10th (ST10) or 7th semester students (ST7)) with 95% confidence intervals (**b**) GM for manual (MP; blue) or electronic probing (EP; green) differentiated by level of examiner’s experience (dentist, 10th (ST10) or 7th semester students (ST7)) with 95% confidence intervals.

**Table 1 diagnostics-12-00042-t001:** Probing pocket depths (PPD) and gingival margins (GM) (*±standard deviations*) measured either with manual probe (MP) or electronic probe (EP), Spearman’s correlation coefficient.

	MP	EP	*p*-Value	Correlation (Spearman’s Rho)
PPD (in mm)	2.82 ± 1.33	2.44 ± 1.35	<0.001 *	0.685 *
**Δ**PPD	0.38			
GM (in mm)	−0.71 ± 1.25	−0.64 ± 1.19	0.195	0.674 *
**Δ**GM	0.07			

* Statistical significance (*p* < 0.05).

**Table 2 diagnostics-12-00042-t002:** Mean probing pocket depths (PPD) and gingival margins (GM) (*± standard deviations)* measured either with manual probe (MP) or electronic probe (EP) differentiated by level of examiner’s experience (dentist, 10th or 7th semester students; ST10, ST7), Spearman’s correlation coefficient.

	Examiner	MP	EP	*p*-Value	Correlation (Spearman’s Rho)
PPD (in mm)	Dentist	2.61 ± 1.33	2.40 ± 1.29	0.002 *	0.810 *
**Δ**PPD		0.21			
PPD (in mm)	ST10	3.31 ± 1.37	2.49 ± 1.35	<0.001 *	0.617 *
**Δ**PPD		0.82			
PPD (in mm)	ST7	2.92 ± 1.19	2.48 ± 1.48	0.195	0.476 *
**Δ**PPD		0.44			
GM (in mm)	Dentist	−0.29 ± 0.70	−0.22 ± 0.70	0.047 *	0.622 *
**Δ**GM		0.07			
GM (in mm)	ST10	−1.47 ± 1.70	−1.24 ± 1.67	0.203	0.699 *
**Δ**GM		0.23			
GM (in mm)	ST7	−1.09 ± 1.40	−1.16 ± 1.23	0.497	0.598 *
**Δ**GM		0.07			

* Statistically significant (*p* < 0.05).

**Table 3 diagnostics-12-00042-t003:** Differences when measuring probing pocket depths (PPD) and gingival margins (GM) between manual probing (MP) and electronic probing (EP) in mm and its following percentages (%) for the observed differences differentiated by level of examiners experience (dentist, 10th or 7th semester students; ST10, ST7).

**PPD (in mm)**	**−7**	**−6**	**−5**	**−4**	**−3**	**−2**	**−1**	**0**	**1**	**2**	**3**	**4**	**5**
Dentist					0.07	0.65	10.34	58.48	28.66	1.29	0.29	0.14	0.07
ST10					0.6	1.0	6.2	31.5	35.6	20.0	4.5	0.4	0.2
ST7			0.2	0.7	2.5	5.0	12.7	25.0	34.6	15.3	3.2	0.7	
**GM (in mm)**	**−7**	**−6**	**−5**	**−4**	**−3**	**−2**	**−1**	**0**	**1**	**2**	**3**	**4**	**5**
Dentist				0.3	0.4	2.5	7.0	85.0	3.1	1.4	0.3	0.1	
ST10	0.4		0.4	0.4	1.9	8.2	20.2	51.1	10.5	5.4	1.0	0.4	
ST7		0.2		0.6	2.2	5.2	14.4	43.5	27.0	4.4	1.7	0.7	

The numbers in bold represent the main statements (differences in mm within groups).

**Table 4 diagnostics-12-00042-t004:** Time needed for manual probing (MP) and electronic probing (EP) (in min) differentiated by level of examiner’s experience (dentist, 10th of 7th semester students, ST10, ST7).

Examiner	MP	EP	*p*-Value
Dentist	14 ± 4	16 ± 4	0.152
ST10	21 ± 10	21 ± 13	0.875
ST7	33 ± 14	37 ± 9	0.500
All	21 ± 11	23 ± 11	0.272

**Table 5 diagnostics-12-00042-t005:** Patients’ pain sensitivity determined by VAS (0 = no pain to 10 = extreme pain) for manual probing (MP) or electronic probing (EP) differentiated by level of examiner’s experience (dentist, 10th of 7th semester students, ST10, ST7).

Examiner	MP	EP	*p*-Value
Dentist	3.47 ± 2.23	2.76 ± 2.32	0.177
ST10	3.68 ± 2.04	1.72 ± 2.26	0.105
ST7	3.80 ± 1.60	6.00 ± 1.41	0.198
All	3.61 ± 2.04	3.31 ± 2.66	0.615

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
