# Peer review of "Clinical Evaluation of a New Electronic Periodontal Probe: A Randomized Controlled Clinical Trial"

_diagnostics, 2021, doi:10.3390/diagnostics12010042_

Round 1

Reviewer 1 Report

Congratulation to your manuscript. It is well organized and clearly written. However, some amendments should be done:

1.) I would prefer to use MP before EP in the sentences/phrases (if the context allows it) to be more consistently in the text/manuscript.

2.) The authors wrote in the lines 220 -221 of page 7 “Paraphrased, a mean difference between both probes could be zero, but could be the result of different errors of e.g. 1 mm in both directions that neutralise each other.”

This is a relevant point and should be looked at more closely. Please show the percentage distribution of the absolute deviation of the measurements (DDP/GM) per examiner (for repeated measurements) in a table.

3.) What do the authors mean with inter-individual reading errors in the conclusion?

  1. Conclusions Standard manual probing and the use of an electronic probe correlated very well, albeit PPD measurements showed a statistically significant difference, which was below a known measuring error of 1mm. Duration for mere measurements and patient acceptance were equal in both probes. Considering inter-individual reading errors using a manual probe, electronic probing can be seen as a practical alternative with advantages that go beyond measurements.

Do the authors refer to the measurement of PPD in general, what they did in the following two phrases?

PPD, defined as the distance between the gingival margin and the bottom of the periodontal pocket is generally measured with conventional manual probes labeled with a millimeter-scale. It is obvious that this method has certain limitations due to inter-individual varying pressures while inserting the probe, different inflammatory conditions of the gingival tissue, wrong angulation of the probe as well as errors while reading the scale or transferring measurements in the dental record [13,14].

(2) reading errors, different angles as well as anatomical depth of the pocket, severity of the disease and smoking habits have an impact on probing independent on whether a manual or electronic probe is used [21,22].

or

Do the authors refer to errors while reading the scale or transferring measurements in the dental record as they described in the lines 56-57 of page 2?

4.) Corrections (red marked) and suggestions (blue marked) regarding the English language can be found in the revised manuscript.

Author Response

Manuscript ID: diagnostics-1470759

Title: Clinical evaluation of a new electronic periodontal probe: A randomized controlled clinical trial

Dear Editors, Dear Reviewer,

we have now carefully considered your helpful annotations  which really helped us to amend the manuscript. The cover letter contains our responses to the concerns as well as any revisions made in the text.

------------------------
Reply to reviewer #1:

  1. Concern of the reviewer:

I would prefer to use MP before EP in the sentences/phrases (if the context allows it) to be more consistently in the text/manuscript.

Our response:

We have changed the order and used MP before EP (also in figure 1 and 2) in order to be unique if the context has allowed it.

Revised text:

Examples given:

In Abstract (page 1, lines 19-22):

This randomized controlled trial compared measurements of a standard manual (MP) with those of an electronic pressure-sensitive periodontal probe (EP) and its influence on patients` acceptance and practicability. In 20 patients (2,436 measuring points) PPD and GM were measured either with MP or EP by…

  1. Concern of the reviewer:

The authors wrote in the lines 220 -221 of page 7 “Paraphrased, a mean difference between both probes could be zero, but could be the result of different errors of e.g. 1 mm in both directions that neutralize each other.”

This is a relevant point and should be looked at more closely. Please show the percentage distribution of the absolute deviation of the measurements (DDP/GM) per examiner (for repeated measurements) in a table.

Our response:

The differences between MP and EP (MP-EP) in mm were calculated. The observed differences were presented in percentages (%) using an additional table (table 3).

Revised text: (page 4 / line150 – page 5 / line 160)

A subanalyis differentiating between dentist, 7th and 10th semester students (ST7, ST10) revealed measurement deviations for both probes (Table 2, Figure 3).  A ΔPPD of 0.21mm in dentist (p=0.002) and of 0.82mm in ST10 (p<0.001) was significantly different, while a ΔPPD of 0.44mm in ST7 was not. The measurement of GM showed a Δ GM of 0.07mm in the dentist and ST7, which was only significant in the dentist (p=0.047) and a ΔGM of 0.23 mm in ST10 which did not reach the level of significance (p=0.203) (Table 2 and Figure 3). Spearman´s rho reached the level of significance in all parameters and examiners. The percentage distribution of the absolute deviation of both (PPD/ GM) in repeated measurements per examiner are presented in table 3. The intra-reader reliabilities (calculating the intra-class correlation coefficient) for the examiners (Dentist, ST10, ST7) for PPD were 0.844, 0.544 and 0.440 and for GM 0.604, 0.729 and 0.572 respectively.

Table 3. Differences when measuring Probing pocket depths (PPD) and gingival margins (GM) between manual probing (MP) and electronic probing (EP) in mm and its following percentages (%) for the observed differences differentiated by level of examiners experience (dentist, 10th or 7th semester students; ST10, ST7)

PPD (in mm)

-7

-6

-5

-4

-3

-2

-1

0

1

2

3

4

5

Dentist

0.07

0.65

10.34

58.48

28.66

1.29

0.29

0.14

0.07

ST10

0.6

1.0

6.2

31.5

35.6

20.0

4.5

0.4

0.2

ST7

0.2

0.7

2.5

5.0

12.7

25.0

34.6

15.3

3.2

0.7

GM (in mm)

-7

-6

-5

-4

-3

-2

-1

0

1

2

3

4

5

Dentist

0.3

0.4

2.5

7.0

85.0

3.1

1.4

0.3

0.1

ST10

0.4

0.4

0.4

1.9

8.2

20.2

51.1

10.5

5.4

1.0

0.4

ST7

0.2

0.6

2.2

5.2

14.4

43.5

27.0

4.4

1.7

0.7

  1. Concern of the reviewer:

What do the authors mean with inter-individual reading errors in the conclusion?

  1. Conclusions: Standard manual probing and the use of an electronic probe correlated very well, albeit PPD measurements showed a statistically significant difference, which was below a known measuring error of 1mm. Duration for mere measurements and patient acceptance were equal in both probes. Considering inter-individual reading errorsusing a manual probe, electronic probing can be seen as a practical alternative with advantages that go beyond measurements.

Our response:

The Electronic probe measures the resistance at the tip-bottom while inserting the tip. This value is automatically calculated into millimeters if a pre-defined resistance is achieved. Due to this standardization EP measurements can be seen as objective.

Manual probes are normally inserted without calibrated pressure (whereas in the present study a pressure calibrated probe was used) and examiners visually read out from a millimeter-scale resulting in ‘inter-individual’, subjective results.

Therefore, we have added “subjective/objective” for a better understanding. Additionally, we picked up the explanation that the standardized resistant force is the key point for objective measurements.

Revised text: (page 9 / line 314 –  317)

Considering subjective inter-examiner reading errors using a manual probe, objective electronic probing on basis of a standardized resistance can be seen as a practical alternative as the need for assistance and time for documentation is saved.

  1. Concern of the reviewer:

Do the authors refer to the measurement of PPD in general, what they did in the following two phrases?

PPD, defined as the distance between the gingival margin and the bottom of the periodontal pocket is generally measured with conventional manual probes labeled with a millimeter-scale. It is obvious that this method has certain limitations due to inter-individual varying pressures while inserting the probe, different inflammatory conditions of the gingival tissue, wrong angulation of the probe as well as errors while reading the scale or transferring measurements in the dental record [13,14].                    (2) reading errors, different angles as well as anatomical depth of the pocket, severity of the disease and smoking habits have an impact on probing independent on whether a manual or electronic probe is used [21,22].

or

Do the authors refer to errors while reading the scale or transferring measurements in the dental record as they described in the lines 56-57 of page 2?

Our response:

Due to the technique of the electronic probe using resistance while inserting the tip and calculating the probing pocket depth, spoken out loud throughout a speaker and transferred into the dental record we referred to both. Due to a better understanding we have added additional information regarding the probe in the introduction part of this manuscript.   

Revised text: (page 2 / line 63 – line 67)

…“PA-ON Parometer” (orangedental, Biberach, Germany) has been introduced to the market (Figure 1). Measurements were taken calculating the insertion depth of a thin steel tip with a plastic cover at a pre-defined resistance at 20 g /0.2 Newton pressure. Due to a computer-based measuring-sequence, tooth-to-tooth data is recorded on a chip, read out loud and transferred wireless in a computer based periodontal-chart.

  1. Concern of the reviewer:
    Corrections (red marked) and suggestions (blue marked) regarding the English language can be found in the revised manuscript.

Our response:

Thank you for helpful corrections and suggestions.

Revised text:

Not applicable

Reviewer 2 Report

Validating a novel diagnostic instrument requires an objective control i.e. what is the true value of a specific parameter. In this study the authors compared mechanical and electronic periodontal probes but were unable to include a "true reference"

While examiners with different levels of experience were included, the authors failed to show inter- and intrarater agreement. Also, it should be added whether or not the Spearman correlations were significant.

While the study itself has been properly conducted and presented, the authors should rewrite and shorten the introduction to better guide the reader.

In addition, a native speaker with proficiency in scientific writing should be able to eliminate phrases such (line 64) "found its way on the market"

Author Response

Manuscript ID: diagnostics-1470759

Title: Clinical evaluation of a new electronic periodontal probe: A randomized controlled clinical trial

Dear Editors, Dear Reviewer,

we have now carefully considered your helpful annotations  which really helped us to amend the manuscript. The cover letter contains our responses to the concerns as well as any revisions made in the text.

------------------------
Reply to reviewer #2:

  1. Concern of the reviewer:

Validating a novel diagnostic instrument requires an objective control i.e. what is the true value of a specific parameter. In this study the authors compared mechanical and electronic periodontal probes but were unable to include a "true reference".

Our response:

Measuring Probing Pocket Depth (PPD) is the prerequisite for diagnostics of periodontal disease. Up to now manual probing is the “golden standard” and so far, the only method to assess the status (and destruction) of the periodontium. Thus, MP must be seen as the “true reference”. (Listgarten et. al. 1976, Listgarten et. al. 1980). For a better understanding, we have also included this information into the introduction section.

Revised text: (page 2 / line 51 – line 56)

PPD measurement is a fundamental prerequisite for diagnostics since it differentiates between the healthy and clinically diseased pocket [14,15]. Up to now, manual probing is the ‘golden standard’ even if this analogue, minimal-invasive method has certain limitations due to inter-individual varying pressures while inserting the probe, different inflammatory conditions of the gingival tissue, wrong angulation of the probe as well as errors while reading the scale or transferring measurements in the dental record [16,17].

  1. Concern of the reviewer:

While examiners with different levels of experience were included, the authors failed to show inter- and intrareader agreement. Also, it should be added whether or not the Spearman correlations were significant.

Our response:

The Spearman correlations (table 1) were significant for PPD (0.685; p<0.001) and GM (0.674; p<0.001).

The same holds for the correlations in the stratified results in table 2. This information has been added with “ * ” in the table and text.  The intra-reader reliabilities (calculating the intra-class correlation coefficient) for the examiners (dentist, 10th semester and 7th semester student) were: PPD: 0.844; 0.544; 0.440 and GM:   0.604; 0.729; 0.572 and added in the text.

Inter-reader agreement cannot be calculated as one patient was only examined by one examiner.

Revised text:

In Abstract: (page 1 / line 27 – line 28)

There was a statistically significant correlation of both probes (Spearman’s rho correlation coefficient GM: 0.674, PPD: 0.685).

In Results: (page 4 / line 141 – page 5 / line 165)

3.1. Measurements of probing pocket depths (PPD) and gingival margins (GM)

In all examiners, mean PPD varied significantly between MP and EP with a mean ΔPPD of 0.38mm (p<0.001), while GM revealed only a ΔGM of 0.07 (p=0.195). However, both probes resulted in a high correlation in both parameters. Spearman’s rho for PPD was 0.685 and for GM: 0.674, both being statistically significant (p<0.001) (Table 1).

Table 1. Probing pocket depths (PPD) and gingival margins (GM) (± standard deviations) measured either with manual probe (MP) or electronic probe (EP), Spearman`s correlation coefficient.

MP

EP

P-value

Correlation

(Spearman’s rho)

PPD (in mm)

2.82 ± 1.33

2.44 ± 1.35

< 0.001*

0.685*

ΔPPD

0.38

GM (in mm)

-0.71 ± 1.25

-0.64 ± 1.19

0.195

0.674*

Δ GM

0.07

*Statistical significance (p < 0.05)

A subanalyis differentiating between dentist, 7th and 10th semester students (ST7, ST10) revealed measurement deviations for both probes (Table 2, Figure 3). A ΔPPD of 0.21mm in dentist (p=0.002) and of 0.82mm in ST10 (p<0.001) was significantly different, while a ΔPPD of 0.44mm in ST7 was not. The measurement of GM showed a Δ GM of 0.07mm in the dentist and ST7, which was only significant in the dentist (p=0.047) and a ΔGM of 0.23 mm in ST10 which did not reach the level of significance (p=0.203) (Table 2 and Figure 3). Spearman´s rho reached the level of significance in all parameters and examiners. The percentage distribution of the absolute deviation of both (PPD/ GM) in repeated measurements per examiner are presented in table 3. The intra-reader reliabilities (calculating the intra-class correlation coefficient) for the examiners (Dentist, ST10, ST7) for PPD were 0.844, 0.544 and 0.440 and for GM 0.604, 0.729 and 0.572 respectively.

Table 2. Mean Probing pocket depths (PPD) and gingival margins (GM) (± standard deviations) measured either with manual probe (MP) or electronic probe (EP) differentiated by level of examiner`s experience (dentist, 10th or 7th semester students; ST10, ST7), Spearman`s correlation coefficient.

Examiner

MP

EP

P-value

Correlation

(Spearman’s rho)

PPD (in mm)

Dentist

2.61 ± 1.33

2.40 ± 1.29

0.002*

0.810*

Δ PPD

0.21

PPD (in mm)

ST10

3.31 ± 1.37

2.49 ± 1.35

<0.001*

0.617*

ΔPPD

0.82

PPD (in mm)

ST7

2.92 ± 1.19

2.48 ± 1.48

0.195

0.476*

  ΔPPD

0.44

GM (in mm)

Dentist

-0.29 ± 0.70

-0.22 ± 0.70

0.047*

0.622*

ΔGM

0.07

GM (in mm)

ST10

-1.47 ± 1.70

-1.24 ± 1.67

0.203

0.699*

Δ GM

0.23

GM (in mm)

ST7

-1.09 ± 1.40

-1.16 ± 1.23

0.497

0.598*

ΔGM

0.07

*Statistically significant (p < 0.05)

In Discussion: (page 7 / line 228 – line 231)

A significant deviation to manual probing could devalue the electronic probe. There are, however, many aspects that should be considered: (1) the overall correlation of both measurements (Spearman’s rho: 0.685*) was high and statistically significant.

  1. Concern of the reviewer:
    While the study itself has been properly conducted and presented, the authors should rewrite and shorten the introduction to better guide the reader.

Our response:

We have rewritten and tried to shorten the introduction (however there were proposals by other reviewers that had to be included).

Revised text: (page 1 / line 34 – page 2 / line 75)

1. Introduction

Periodontal disease is a widespread infectious disease of tooth supporting tissues with a prevalence of more than 50% [1]. It causes a significant economic burden both in the US and in Europe [2]. Chronic inflammation with progressive alveolar bone loss [3] does not only result in tooth-loss [4], there are also clinical and inflammatory relationships with other chronic metabolic, inflammatory, and vascular diseases, such as diabetes [5], cardiovascular diseases [6], chronic obstruction pulmonary disease (COPD) [7], metabolic syndrome, and obesity [8,9]. Therefore, early diagnosis and prevention gains more and more importance.

According to the new classification of periodontal diseases [10] as well as the EFP S3 level guideline for its treatment [11], exact measurements of periodontal parameters are necessary for adequate diagnostics, monitoring treatment-outcomes and the decision of further treatment-options.Probing pocket depth (PPD), defined as the distance between the gingival margin and the bottom of the periodontal pocket, together with clinical attachment level-loss (CAL-loss) and bone-loss as well as bleeding on probing (BOP) are clinical signs of periodontal destruction [12]. Only in-time diagnosis followed by adequate therapy can guarantee effective treatment [13].

PPD measurement is a fundamental prerequisite for diagnostics since it differentiates between the healthy and clinically diseased pocket [14,15]. Up to now, manual probing is the ‘golden standard’ even if this analogue, minimal-invasive method has certain limitations due to inter-individual varying pressures while inserting the probe, different inflammatory conditions of the gingival tissue, wrong angulation of the probe as well as errors while reading the scale or transferring measurements in the dental record [16,17].

In order to minimize errors resulting from differing pressure-forces, periodontal probes with a pressure calibration of 20g/0.2 Newton have been developed [18]. However, they are very susceptible to defects, much more expensive and its hygienic preparation is very difficult. First electronic pressure sensitive probes did not bring any advantage, as they coronally penetrate the junctional epithelium [19] and patients` acceptance was low.

In recent years, a new, computer-based, electronic and pressure-calibrated probe (PA-ON Parometer, orangedental, Biberach, Germany) has been introduced to the market (Figure 1). Measurements are taken calculating the insertion depth of a thin steel tip with a plastic cover at 20 g/0.2 Newton pressure using resistance. Due to a computer-based measuring-sequence, tooth-to-tooth data is recorded on a chip, read out loud and transferred wireless in a computer based periodontal-chart.

It was the aim of this randomized, controlled, clinical trial to compare this electronic with a conventional and established manual probe on gaining reliable data and its accordance (primary outcome). In a sub-analysis it should be examined if clinical experience of the examiner had an influence on measurements. Time needed for data assembling as well as patients` acceptance while probing were further outcomes.

It is hypothesized that periodontal diagnostics by electronic is at least comparable to established manual probing. Furthermore, it was questioned if electronic probing could be a faster alternative and being preferred by patients.

  1. Concern of the reviewer:
    In addition, a native speaker with proficiency in scientific writing should be able to eliminate phrases such (line 64) "found its way on the market"

Our response:

We have edited the English language.

Revised text

Example given: (page 2 / line 62 – line 64)

In recent years a new, computer-based, electronic and pressure-calibrated probe, the “PA-ON Parometer” (orangedental, Biberach, Germany) has been introduced to the market (Figure 1).

Reviewer 3 Report

This randomized clinical trial compared an electronic probe with a conventional manual probe concerning reliability, accordance, the influence of experience of the examine, time required and ‘quality of life’ of the patients.

In the Introduction, the authors present well the rationale why this study is needed. I would only recommend to add some socioeconomic background on the economic burden of periodontal disease. For example (Listl et al. 2015 and Botelho et al. 2021).

This study presents a robust methodology however with three concerns:

1. When the authors refer to "quality of life" in the secondary outcomes, I would expect a validated questionnaire to this purpose. Yet, authors did mention a questionnaire without mentioning which.

2. Authors have only explained the VAS for the examiners satisfaction in the Results methods. Please replace this into the M&M section.

3. Why have did you not add standard deviation values into the text? Any reason?

In the Conclusion:

"Considering inter-individual reading errors using a manual probe, electronic probing can be seen as a practical alternative with advantages that go beyond measurements." This sentence is poorly objective, and may lead to subjective interpretation. Please rephrase or be more concise on the advantages.

Author Response

Manuscript ID: diagnostics-1470759

Title: Clinical evaluation of a new electronic periodontal probe: A randomized controlled clinical trial

Dear Editors, Dear Reviewer,

we have now carefully considered your helpful annotations  which really helped us to amend the manuscript. The cover letter contains our responses to the concerns as well as any revisions made in the text.

------------------------
Reply to reviewer #3:

  1. Concern of the reviewer:
    In the Introduction, the authors present well the rationale why this study is needed. I would only recommend to add some socioeconomic background on the economic burden of periodontal disease. For example (Listl et al. 2015 and Botelho et al. 2021).

Our response:

We included the recent Botelho et al. (2021) study in the introduction.

Revised text: (page 1 / line 35 – line 36)

          Periodontal disease is a widespread infectious disease of tooth supporting tissues with a prevalence of more than 50% [1].  It causes a significant economic burden both in the US and in Europe [2].

  1. Concern of the reviewer:
    This study presents a robust methodology however with three concerns:

When the authors refer to "quality of life" in the secondary outcomes, I would expect a validated questionnaire to this purpose. Yet, authors did mention a questionnaire without mentioning which.

Our response:

We used an independently developed questionnaire in the style of the Quality of life (QoL) and added this missing information.

Revised text: (page 4 / line 125 – line 126)

  • The subjective experience of the patients was determined by an independently developed questionnaire in the style of Quality of life (QoL).
  1. Concern of the reviewer:

 Authors have only explained the VAS for the examiners` satisfaction in the Results methods. Please replace this into the M&M section.

Our response:

We unified the order in the M&M and Result sections

Revised text:  (page 4 / line 119 – line 128 and page 6 / line 184 – page 7 line 203)

2.5. Further parameters:

  • Time needed to complete periodontal charting was counted for each method and each examiner and measured in minutes using a stopwatch.
  • Questionnaires by means of visual analogue scale (VAS) were used to gain information concerning patient’s pain-sensation (0 [no pain] to 10 [extreme pain]) during probing.
  • The subjective experience of the patients was determined by an independently developed questionnaire in the style of Quality of life (QoL).
  • An evaluation by dental professionals was performed in order to determine expectations and satisfaction (VAS) with MP compared to E

3.3. Time needed for measurements

The absolute EP measuring time (23 ± 11 minutes) was in average 2 minutes faster, but was not a significant timesaver compared with MP (21 ± 11 minutes) (p<0.05). Time needed decreased with examiners experience (Table 4).

3.4. Patients pain sensitivity (VAS)

The comparison of VAS pain did not result in statistically significant differences neither dependent on examiners experience nor on probe used (MP: 3.61± 2.04, EP: 3.31± 2.66) (Table 5).

3.5. Patients` subjective experience

Nearly all participants (19 of 20) had not been familiar with EP before and their expectations varied from easier (8), more pleasant (10), more precise (5) to no expectations (7). While the evaluation of EP in comparison to MP revealed ‘much better’ (4), ‘better’ (6) ‘worse’ (5) and ‘no difference’ (6), most of them would prefer EP (12 EP; 6 MP) for future measurements.

3.6. Evaluation by dental professionals

All examiners (ST7, ST10, dentist) considered EP to be faster, more precise and easier. The VAS-values for satisfaction with the clinical use (1=very satisfied and 10= not satisfied)  showed a moderate satisfaction in dentists (5.11), in ST7 (5.54), while ST10 graded 3.06.

  1. Concern of the reviewer:

 Why have did you not add standard deviation values into the text? Any reason?

Our response:

Standard deviations have been added.

Revised text:

Examples given: (page 6 / line 188 – line 191)

3.3. Time needed for measurements

The absolute EP measuring time (23 ± 11 minutes) was in average 2 minutes faster, but was not a significant timesaver compared with MP (21 ± 11 minutes) (p<0.05). Time needed decreased with examiners experience (Table 4).

3.4. Patients pain sensitivity (VAS)

The comparison of VAS pain sensitivity did not result in statistically significant differences neither dependent on examiners experience nor on probe used (MP: 3.61± 2.04, EP: 3.31± 2.66) (Table 5).

  1. Concern of the reviewer:
    In the Conclusion:

"Considering inter-individual reading errors using a manual probe, electronic probing can be seen as a practical alternative with advantages that go beyond measurements." This sentence is poorly objective, and may lead to subjective interpretation. Please rephrase or be more concise on the advantages.

Our response:

After rewriting the introduction, we have also adapted the conclusion in order to mention the advantages of the electronic probe

Revised text: (page 9 / line 310 – line 317)

5. Conclusions

Standard manual probing and the use of an electronic probe correlate very well, albeit PPD measurements showed a statistically significant difference, which was below a known measuring error of 1mm. Duration for pure measurements and patient acceptance were comparable in both probes. Considering subjective inter-examiner reading errors using a manual probe, objective electronic probing on basis of a standardized resistance can be seen as a practical alternative as the need for assistance and time for documentation is saved.

Round 2

Reviewer 2 Report

Thanks for addressing my comments

Author Response

Thank you for your helpful review.